# Neonatal Hypoxic-Ischemic Encephalopathy Grading from Multi-Channel EEG Time-Series Data Using a Fully Convolutional Neural Network

Shuwen Yu [1,2,*], William P. Marnane [1,2], Geraldine B. Boylan [2,3] and Gordon Lightbody [1,2,*]

1 Electrical and Electronic Engineering, University College Cork, T12 YF78 Cork, Ireland; l.marnane@ucc.ie
2 INFANT Research Centre, University College Cork, T12 DC4A Cork, Ireland; g.boylan@ucc.ie
3 Pediatrics and Child Health, University College Cork, T12 DC4A Cork, Ireland
* Correspondence: 120220809@umail.ucc.ie (S.Y.); g.lightbody@ucc.ie (G.L.)

**Abstract:** A deep learning classifier is proposed for grading hypoxic-ischemic encephalopathy (HIE) in neonates. Rather than using handcrafted features, this architecture can be fed with raw EEG. Fully convolutional layers were adopted both in the feature extraction and classification blocks, which makes this architecture simpler, and deeper, but with fewer parameters. Here, two large (335 h and 338 h, respectively) multi-center neonatal continuous EEG datasets were used for training and testing. The model was trained based on weak labels and channel independence. A majority vote method was used for the post-processing of the classifier results (across time and channels) to increase the robustness of the prediction. A dimension reduction tool, UMAP, was used to visualize the model classification effect. The proposed system achieved an accuracy of 86.09% (95% confidence interval: 82.41–89.78%), an MCC of 0.7691, and an AUC of 86.23% on the large unseen test set. Two convolutional neural network architectures which utilized time-frequency distribution features were selected as the baseline as they had been developed or tested on the same datasets. A relative improvement of 23.65% in test accuracy was obtained as compared with the best baseline. In addition, if only one channel was available, the test accuracy was only reduced by 2.63–5.91% compared with making decisions based on the eight channels.

**Keywords:** hypoxic-ischemic encephalopathy (HIE); EEG; fully convolutional neural network; UMAP

## 1. Introduction

Hypoxic-ischemic encephalopathy (HIE) is caused by a lack of oxygen and blood supply to the neonatal brain around the time of birth [1,2]. The severity of this injury can be assessed clinically, soon after birth, as mild, moderate, or severe. However, clinical assessment can be subjective and also difficult to perform if the infant requires sedation. In the most severe cases, the baby may not survive the injury, and those that do survive may have permanent disabilities, such as epilepsy, cerebral palsy, and learning difficulties [3,4]. Therapeutic hypothermia is a standard method to treat neonates with moderate or severe HIE—the neonate is cooled to 33.5 °C for 72 h [5–7]. However, for this treatment to be effective, it ideally should be commenced within 6 h after birth [8,9]. Hence, an accurate assessment of the severity of encephalopathy is required soon after birth to instigate therapeutic hypothermia within this narrow window of time.

Electroencephalography (EEG) is an effective, portable, and non-invasive tool to monitor neonatal cortical activity. The severity of the HIE injury can be graded from the neonatal EEG [10]. However, the interpretation of EEG is not straightforward, requiring the expertise of a neurophysiologist, which may not be available in many Neonatal Intensive Care Units (NICU), especially on a 24/7 basis. An automated method is therefore attractive as a solution to provide some estimate of the severity of the HIE injury from the EEG within

the narrow 6 h window of opportunity for the initiation of therapeutic hypothermia [11]. This could also be available as a 24/7 cot-side service.

In the expert visual interpretation of HIE injury, the characteristics of the background EEG such as amplitude, frequency, discontinuity, and sleep–wake cycling [12–14] are analyzed and the resulting grade is determined. In [15,16], some quantitative EEG features were investigated and selected from the amplitude, frequency, and continuity characteristics of the EEG to reproduce the process of visual HIE grading. Some machine learning methods have been developed to automatically grade the HIE, with the early techniques focusing on grade classification based on quantitative features, which, in some ways, mimics the expert visual interpretation. In particular, Stevenson et al. [17] extracted the amplitude modulation and instantaneous frequency features from the time-frequency distribution extracted over short 64 s epochs. The distribution of these features over a 1 h window provided the feature vector for a multiple linear discriminant classifier. This achieved a classification accuracy of 83% on a relatively small dataset consisting of 62 h of eight-channel EEG from 54 neonates. Ahmed et al. [18] utilized a feature set consisting of 55 basic descriptive features from the frequency, time, and information theory domains, which were extracted from short 8 s windows of EEG. In comparison to [17], this work utilized generic simple features and focused primarily on the use of a much more sophisticated dynamic classifier (the SVM supervector approach). This provided a grade every 80 s that was determined over a sequence of twenty 8 s windows (with 50% overlap). The final EEG grade for the neonate was then determined over an hour of multi-channel EEG. This approach yielded a significant improvement in grading performance over [17], with an accuracy of 87% achieved over the same small development dataset.

An alternative approach that does not depend on either finely crafted features [17] or a wide range of descriptive features [18] is to shift from classical machine learning to deep learning, where the features can be automatically extracted from the data. In recent years, the convolutional neural network has dominated the field of computer vision. Ruarale et al. [19] extended the work of [17], where, instead of manually extracting features from time-frequency analysis, they proposed that the time-frequency distribution feature map could be treated as an image that could be input to a convolutional neural network trained to classify HIE grades. Three parallel convolutional channels automatically extracted features in the direction of time, frequency, and time-frequency from a two-dimensional time-frequency distribution feature map. The same architecture was then also used in [20] and trained with a large dataset, with the final softmax layer substituted by a regression layer (to reflect the ordinal nature of the grades). This approach achieved a classification accuracy of 82.8% on a large dataset in mismatched conditions (they were recorded at different times, centers, machines, sampling frequencies, and even protocols).

Rather than using any features of the EEG as input, some convolutional neural network architectures have been successfully applied to detect neonatal seizures and sleep stage detection directly from the raw EEG signal, with minimal preprocessing. A fully convolutional neural network (FCN) architecture was presented in [21] and successfully solved the problem of neonatal seizure detection from the raw EEG signal. HIE grading requires a much wider receptive field than seizure detection, in order to capture the sort of changes in the EEG signal that may occur over say 80 s as opposed to 10 s for neonatal seizure detection. In [22], a convolutional neural network (CNN) architecture was used for sleep state detection with raw EEG, which achieved the highest classification accuracy both in preterm and term neonates.

Unlike seizure detection, where a seizure event in the EEG can be marked with a 1-s resolution, each long segment (typically 1 h) of EEG in the HIE grading problem, has only one HIE grade annotation. It is tempting therefore to address this problem as a sequence classifier using an architecture such as the transformer, which is suited to long sequence information processing [23]. Recently, some transformer models, such as vision transformer and swin transformer, have achieved very good performance in many tasks and have outperformed convolutional neural networks. Recently, however, there have been some

notable improvements in convolutional networks, such as ConvNets [24], which have achieved better performance than the swin transformer in vision tasks and maintained the simplicity and efficiency of the convolutional network.

In this paper, we used a fully convolutional neural network with raw EEG to classify the HIE grades. Without the fully connected layers in the classifier block, this network has much fewer parameters and also maintains the temporal position information. This network was trained on a large dataset of continuous unedited multi-channel EEG from a multi-center study. The performance of the HIE grading network was then demonstrated on another large unseen and unedited multi-center dataset, under mismatched conditions.

## 2. Method

### 2.1. Dataset

This is a secondary data analysis from newborn infants recruited between January 2011 to February 2017 as part of two multicentre cohort studies across eight European tertiary neonatal intensive care units. Newborn infants born after 36+0 weeks gestational age requiring EEG monitoring for clinical reasons were eligible for inclusion in the main studies (Algorithm for Neonatal Seizure recognition ANSeR). The main results of the two studies have already been published and the study protocols are available on ClinicalTrials.gov (Identifier: NCT02160171 and NCT02431780). In total, 472 neonates were recruited, but for the purpose of our analysis, only 284 infants who were diagnosed with HIE were selected.

Of these 284 HIE neonates, 35 infants were subsequently excluded, and 68 were held out for future validation. For the remaining 181 neonates, EEG epochs each lasting exactly 1 h were extracted from the EEG records at postnatal ages of 6, 12, 24, 36, and 48 h (if available). The ANSeR1 dataset provided 338 h of multi-channel EEG from 91 neonates across the 6 centers. The ANSeR2 dataset provided 315 h of multi-channel EEG from 90 infants across 8 centers. Compared with other EEG datasets — SIMKAP (48 subjects) for mental workload classification [25], the SEED emotion recognition task dataset (15 subjects, 45 h) [26], and the DEAP dataset (32 subjects, 21.3 h) [27]—our dataset is sizable.

All neonates were recorded with cEEG (Nihon Konhden Neurofax, EEG-1200, Tokyo, Japan or NicoletOne ICU Monitor & Xltek, Natus, Middleton, WI, USA) with a sampling frequency of either 256 Hz or 250 Hz. The electrodes were positioned according to the international 10:20 system with disposable electrodes at F3, F4, C3, C4, T3, T4, Cz, O1/P3, and O2/P4.

Each 1 h segment of multi-channel EEG was graded by an expert neurophysiologist for the severity of the HIE injury using 5 grades: normal, mild, moderate, major, and inactive. In this work, the normal and mild grades were combined as grade 1; this reduces the number of grades to 4. Figure 1 presents clear EEG examples of different HIE grades. The inter-burst interval (IBI) is one of the key characteristics, which makes the EEG discontinuous and is examined by experts in grading the HIE injury from EEG [28]. An inter-burst interval is a period of suppressed EEG, lasting more than 2 s in duration and with an amplitude less than 25 μV peak to peak, sandwiched between larger amplitude bursts of EEG [29]. It can be appreciated from Figure 2 that the ANSeR1 and ANSeR2 datasets have similar distributions for the HIE grades. The classes are, however, substantially unbalanced (this reflects the distribution in a typical NICU), with grade 1 accounting for over 50% of the annotated 1 h long epochs.

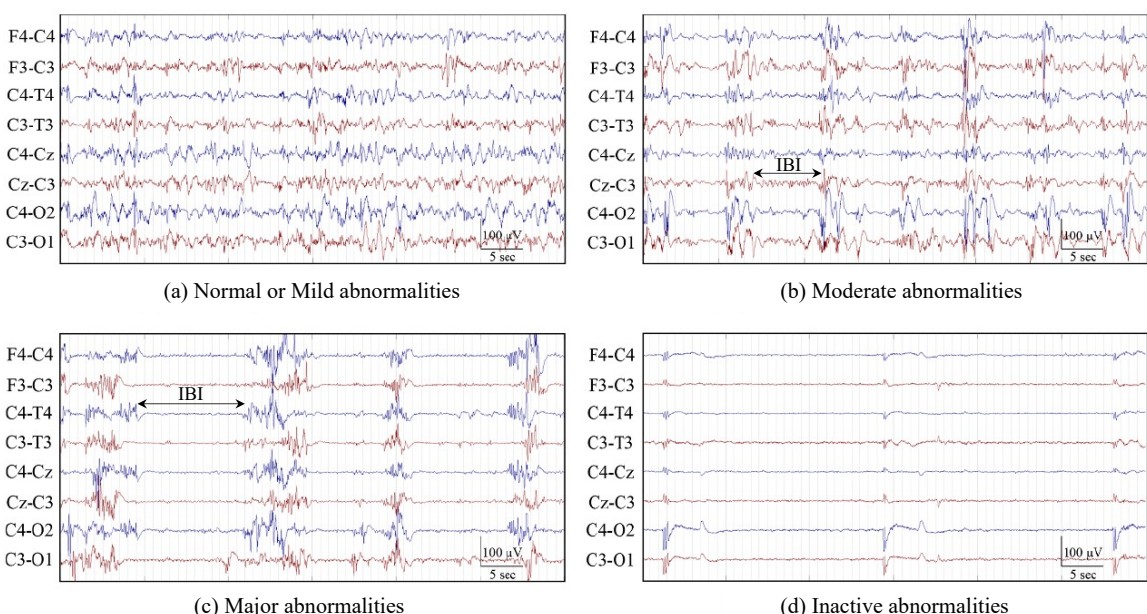

Figure 1. EEG examples for different HIE grades. (**a**) Normal or mild abnormalities (grades 1); (**b**) moderate abnormalities (grade 2); (**c**) major abnormalities (grade 3); (**d**) inactive abnormalities (grade 4). Grades 2 and 3 also indicate the inter-burst intervals (IBI) annotation. (From [19]).

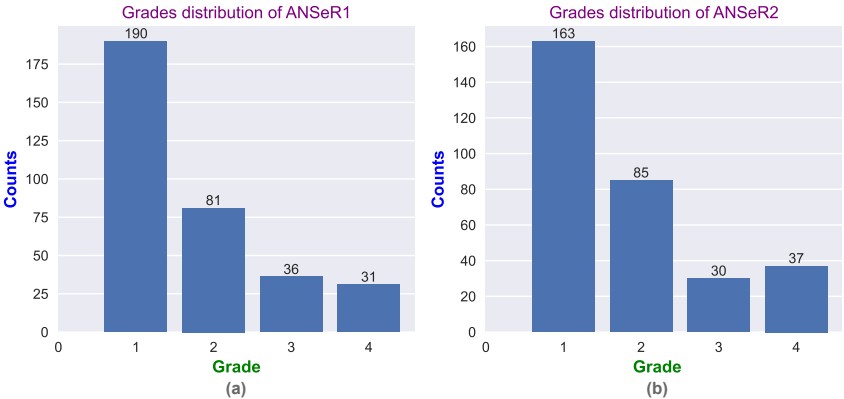

**Figure 2.** The distribution of the HIE grades for (**a**) the ANSeR1 HIE dataset and (**b**) the ANSeR2 HIE dataset.

### 2.2. Fully Convolutional Neural Network

#### 2.2.1. Pre-Processing

A bipolar montage was used with 8 EEG channels: F4-C4, C4-O2/C4-P4, F3-C3, C3-O1/C3-P3, T4-C4, C4-Cz, Cz-C3, and C3-T3. Since the key information in neonatal EEG is located below 13 Hz, a bandpass filter (0.5–12.8 Hz) was employed; the 0.5 Hz cut-off frequency was adopted to avoid very low frequencies such as DC drift. The EEG was then resampled from 256 or 250 Hz to 32 Hz. No other artefact removal procedures were used. It is common in EEG-based classification tasks to reduce the effect of artefacts on performance, using techniques such as independent component analysis to separate the artefacts from the EEG [30]. However, in deep learning, with enough training data, the key features can be learned even from noisy EEG data when artefacts are present. In this work, we are keen to demonstrate that excellent performance can be achieved even when there is minimal preprocessing of the raw EEG.

As HIE is a global injury, all the channels should display EEG characteristics in a similar way. Thus, channel-independent EEG epochs could be used for training a single channel-independent classifier. This allows the number of training samples to increase

by a factor of 8. In keeping with the approach proposed by [17,18] each 1 h graded EEG segment was split into 60 s epochs using a 60 s moving window with 50% of overlap; this was deemed long enough to capture the expected variation in the inter-burst interval (IBI) across grades. Each window was labeled with the grade assigned to that particular 1 h epoch. Technically, each 60 s window has what is called a "weak label", as the actual grade may possibly vary over the one hour.

Because the bandpass filter removed the DC drift, the mean of each channel in the bipolar montage was zero. The standard derivation of each channel across all the training data ranged from $1 \times 10^{-6}$ V $- 1 \times 10^{-4}$ V. In a very basic normalization, all EEG channels were simply scaled up by a factor of $1 \times 10^4$, as input samples which are very small might cause difficulty in the training of the initial convolutional layers which are situated before the first batch normalization layer.

### 2.2.2. FCN Architecture

A one-dimensional fully convolutional neural network (FCN) was used for the HIE grading. The FCN10/13/16 model architectures can be seen in Figure 3; these were composed of 10/13/16 convolutional layers, respectively. The input of the model was a 60 s raw EEG segment sampled at 32 Hz. These 1920 samples can be thought of as an image with a single feature channel, i.e., the input shape is (1,1920).

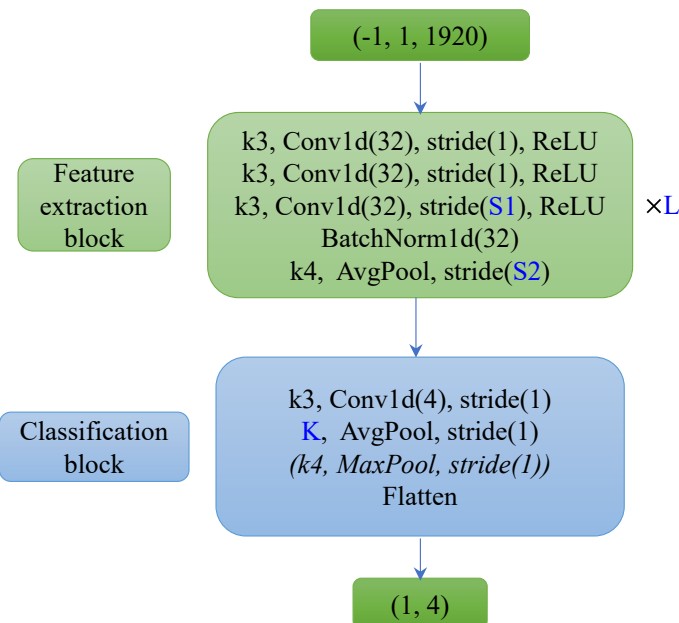

**Figure 3.** Architecture of FCN10/13/16. FCN10 was composed of 3 (L) feature extraction blocks with 32 channels of convolutional layers (which projects the one feature channel of raw EEG segment (1,1920) as 32-channel features) and a classification block with 4 channels of a convolutional layer. FCN13/16 has 4/5 (L) feature extraction blocks with 32 channels of convolutional layers (S1 stride of 1) and a classification block without a MaxPool layer.

Table 1 presents the configuration of the different model parameters, where L is the number of layers of the feature extraction block, and S1 and S2 are the strides of the third convolutional layer and the Average Pooling layer in the feature extraction blocks, respectively. K is the kernel size of the average pooling layer in the classification block. Only the FCN10 has a Max Pooling layer (this helps the feature map size reduce to one within the limited number of layers). The deeper model has a smaller stride and kernel size, which maintains a gradual reduction of the feature map size from the network input to its output.

**Table 1.** Model parameter of different architectures. L is the number of the feature extraction blocks; S1 and S2 are the stride of the third convolutional layer and average pooling layer in the feature extraction blocks. K is the kernel size of the max pooling layer in the classification block.

|       | L | S1 | S2 | K |
|-------|---|----|----|---|
| FCN10 | 3 | 2  | 3  | 3 |
| FCN13 | 4 | 1  | 4  | 3 |
| FCN16 | 5 | 1  | 3  | 2 |

L layers of identical feature extraction blocks were used to learn the features from the raw EEG segments. Each of these feature extraction blocks consists of three 1D convolutional layers with 32 channels. The BatchNorm layer normalizes the learned features and maintains them within a suitable distribution, hence, helping the training process. BatchNorm is mainly implemented by Equation (1). In the training process, for each mini-batch, the moving average mean $\widehat{\mu}_B$ and variance $\widehat{\sigma}_B$ are calculated based on this mini-batch input $x$. This moving average mean and variance are used as the distribution estimation of the whole dataset, as the real distribution of the whole dataset cannot be acquired during the training process. Moreover, the learnable scaler and shift $(\gamma, \beta)$ help the model learn a more suitable distribution rather than the fixed ones $(0, 1)$. For deep network training, a wide range of values might occur over the layers from the input to output. This will slow down the training and a learning rate adjustment might be necessary [31]. BatchNorm solves this problem by controlling the data distribution. In addition, batch normalization also plays a regularization role, as the estimated mean and variance on each mini-batch will include noise during the training process, which can mitigate against overfitting.

$$BN(x) = \gamma \odot \frac{x - \widehat{\mu}_B}{\widehat{\sigma}_B} + \beta \tag{1}$$

The pooling layer not only reduces the sensitivity of the convolutional layer to positional information and increases the robustness of the model, but it also plays a downsampling role; this is useful in controlling the receptive field. In the classifier block, the convolutional layer projects from 32 channels down to 4 channels and summarizes the characteristics of the 4 HIE grades. Compared with the usual fully connected classifier layers, these convolutional layers ensure that the model has far fewer parameters, is simpler in structure, and is less likely to overfit. The fully convolutional structure means that decisions made at the output can be traced back through the network to a particular window in the time of the EEG.

2.2.3. Post-Processing

For the validation and test set, a post-processing procedure was implemented to yield the final 1 h epoch prediction result. This procedure was based on a majority vote, which mimicked the clinical annotation of HIE from the EEG—the grade is not constant during the one hour and the final decision is determined by the most frequent grade. In addition, it also improved the robustness of the prediction. This procedure can be seen in Figure 4. Each 60 s segment in each channel is classified as grade 1–4, independently using a single common FCN classifier. The grades are then collated for the 119 (60 s) segments for each of the 8 EEG channels. The resultant HIE grade for each 1 h epoch is then determined by the majority vote (across channels and time).

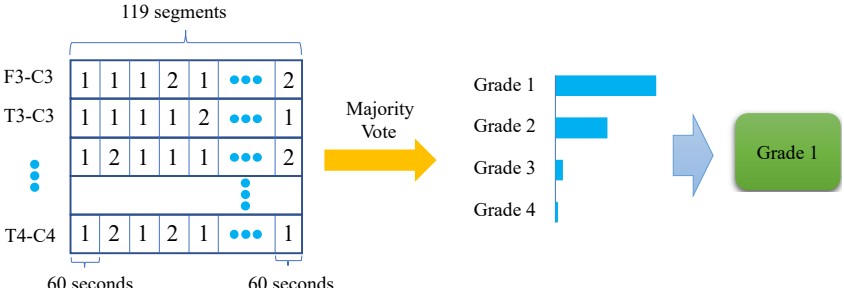

**Figure 4.** Majority vote procedure—HIE grade estimated over 1 h and multiple EEG channels.

### 2.2.4. Metrics

In our work, the prediction accuracy and the area under the receiver-operating curve (AUC-ROC) were used for the primary metrics of the classification performance. In HIE grading tasks, almost all of the published works have reported classification accuracy. However, due to the unbalanced distribution of each class, the classification accuracy cannot sufficiently represent the model performance. AUC measures the ability of a model to correctly classify positive and negative classes, even if one of the classes is more dominant than the other and represents the overall quality of the model performance for different thresholds. The Matthews Correlation Coefficient (MCC) is another common metric in classification tasks which can also represent the classification performance regardless of the number of samples in each class. Compared with accuracy, MCC takes into account the 4 values of the confusion matrix; thus, even if the positive class and negative class are swapped, its value does not change. It can be calculated by Equation (2):

$$MCC = \frac{TP \times TN - FP \times FN}{\sqrt{(TP+FP)(TP+FN)(TN+FP)(TN+FN)}} \tag{2}$$

where *TP* and *TN* are the true positive and the true negative in the confusion matrix, respectively; *FP* and *FN* are the false positive and the false negative, respectively.

The confidence interval (CI) provides a bound of the uncertain estimation. In our work, a 95% confidence interval was combined with the test accuracy, which not only provided a better measure of model performance but also facilitated a fair comparison between models.

### 2.2.5. Visualization

For explainable AI (XAI), it is important to visualize how the CNN can extract features from the EEG. UMAP (Uniform Manifold Approximation and Projection) is a powerful dimension reduction tool for both visualization and pre-processing in machine learning tasks [32]. It is extremely useful to visualize how high-dimensional data looks when mapped onto 2 or 3 dimensions. UMAP constructs a weighted graph representation of the high-dimensional data. The edge weight represents the likelihood of the connection between two points, which was determined by whether their extended radii overlapped. A good radius is very difficult to choose, as too small a value yields too many isolated points, and too large a value causes everything to be connected. Rather than using a fixed radius, UMAP chooses a radius locally for each point, based on the n nearest neighbor. Subsequently, UMAP optimizes a low-dimensional graph using the cross-entropy function to keep them as similar as possible. Compared with another popular dimension reduction technique, namely, t-SNE, UMAP is better at retaining the global structure of the data and is much faster to compute [33,34]. Furthermore, UMAP also outperforms t-SNE when the embedding dimension is more than 2, because it does not need a global normalization [33]. It is particularly useful for generating a low-dimensional representation for further machine learning tasks, such as clustering and classification.

## 3. Results

In our experiments, the ANSeR2 dataset was used for training and the unseen ANSeR1 dataset was used for testing. In many EEG-based classification tasks, the individual model's performance is better than that of a general model [35]. Furthermore, cross-dataset evaluation is more challenging as many subjects may have different environments, devices, and protocols [36]. An early stopping mechanism was adopted, in which a relatively large validation set was selected (for internal cross-validation) from the ANSeR2 training set (20% of epochs) in order to better evaluate the performance of the different models. The significantly large unseen test dataset in ANSeR1 presents a demanding assessment of the generalization ability of the trained classifier. The training set includes 251 h of EEG. The validation set and test set have 63 h and 338 h of EEG, respectively.

The multi-class cross-entropy function was used as the loss function, which can be seen in Equation (3). Here, $\hat{\mathbf{y}}$ is the predicted probability vector of the n classes, and $\mathbf{y}$ is the label vector in one-hot format. It has been noted that the $L_2$ regularization is not identical to weight decay in adaptive gradient algorithms, and it is not effective in Adam. In this work, AdamW is used, which improves the model generalization ability by decoupling the weight decay factor from the optimization. It also makes the hyperparameter tuning easier [37]. The learning rate and weight decay are around $1 \times 10^{-5}$ and 0.1, respectively (these parameters vary slightly across the trained models).

$$l(\mathbf{y}, \hat{\mathbf{y}}) = -\sum_{i=1}^{n} y_i log \hat{y}_i \tag{3}$$

In the training process, eight different networks were trained for each classifier, each with a different random initialization. For each network initialization, the best model was selected in training based on the highest moving average validation AUC. The model performance for each of the eight trained networks was then reported on the test set. Figure 5 shows the test accuracy distribution on the ANSeR1 dataset from these eight trained networks. Three different architectures were compared in this manner. The FCN16 architecture achieved the best performance with a higher mean and smaller spread. The FCN10 network performed better than the FCN13 (with a larger mean, median, maximum, and minimum), but with a significant outlier.

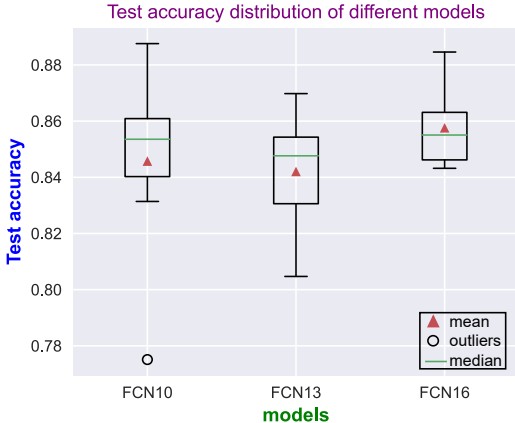

**Figure 5.** Distributions of the test accuracy results (on the ANSeR1 dataset) for each of the network models. For each model, 8 networks were trained, each with a different random initialization.

To reduce the bias of the reported model performance, the predictions provided by the eight trained FCN16 networks were ensembled using a simple majority vote across networks. Table 2 presents the confusion matrix obtained using the ensemble of eight FCN16 model prediction results from the 338 one-hour HIE grades (the ANSeR1 test set). The sensitivity (true positive rate) and the specificity (true negative rate) indicates how likely the positive class and the negative class (the other three classes were treated as the

negative class) was correctly classified. The positive predictive rate (PPV) measures how confident an example was predicted as positive. For example, the sensitivity of class 1 was calculated by TP/(TP + FN) = 181/(181 + 9) = 0.953, the specificity was calculated by TN/(TN + FP) = 124/(124 + 24) = 0.838, and the PPV of class 1 by TP/(TP + FP) = 181/(181 + 24) = 0.883. The sensitivity of grade 1 (0.953) and the PPV of grade 4 (0.963) achieved the highest values, whereas the PPV of grade 3 (0.762) and sensitivity of grade 2 (0.642) have the lowest values, respectively. No epoch was wrongly predicted by more than one grade from the expert-labeled grades. An 86.09% test accuracy from the ensembled prediction was achieved with a 95% CI of 82.41 89.78% and 0.7691 of MCC. One of the FCN16 model prediction ROC curves on the test set can be seen in Figure 6, which yielded an average test AUC of 86.3%. The learning curve can be seen in Figure 7.

**Table 2.** Confusion matrix of 8 ensembled FCN16 models prediction results over the ANSeR1 test set based on the majority vote.

| | | Prediction | | | | | |
| --- | --- | --- | --- | --- | --- | --- | --- |
| | | 1 | 2 | 3 | 4 | sensitivity | specificity |
| True Label | 1 | 181 | 9 | 0 | 0 | 0.953 | 0.838 |
| | 2 | 24 | 52 | 5 | 0 | 0.642 | 0.953 |
| | 3 | 0 | 3 | 32 | 1 | 0.889 | 0.967 |
| | 4 | 0 | 0 | 5 | 26 | 0.839 | 0.997 |
| | PPV | 0.883 | 0.813 | 0.762 | 0.963 | | |

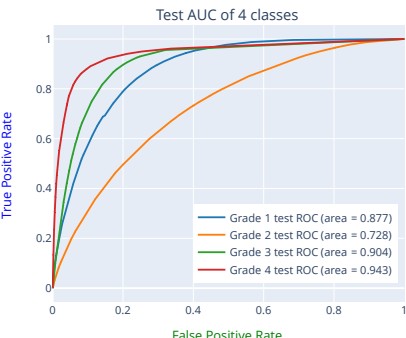

**Figure 6.** ROC curve on the test set from one of the FCN16 networks predictions. The AUC of each class was obtained by treating it as a binary classification problem with the other 3 classes representing the negative class.

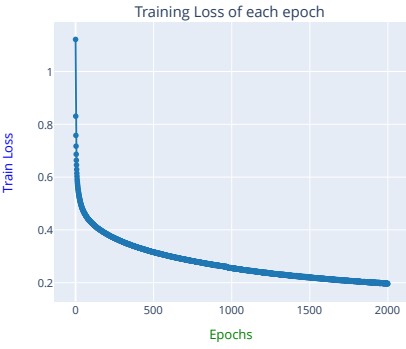

**Figure 7.** Learning curve from one of the FCN16 networks training (approximately 12 h were required for 2000 epochs training on a Tesla V100 GPU).

## 4. Discussion

### 4.1. Comparison with CNN Baseline

Many of the previous works were trained and tested on different datasets; thus, it is difficult to directly compare their performance. Two of them, however, were selected here as the baseline. In [19], a complex convolutional neural network extracted features from the quadratic time-frequency distribution (TFD-CNN). Although it was trained on a small dataset (54 neonates), it was also tested on the ANSeR1 dataset. The same architecture was also employed in [20], but the softmax layer was replaced as a regression layer (TFD-CNN-Reg). The whole ANSeR1&2 dataset was used for development, and a 10-fold cross-validation method was adopted. Therefore, a more direct comparison can be implemented based on this work.

Table 3 shows the comparison between the two baselines and the proposed method configurations. Table 4 presents the performance of the different methods. The proposed FCN16 architecture achieved the best performance (average over eight networks; each with random initialization). Not surprisingly, the TFD-CNN method achieved the lowest performance, due in some ways to a smaller development dataset (62 h). However, even with a substantially larger training set, this architecture with a slight modification of the final layer (TFD-CNN-Reg) only obtained a test accuracy of 82.8%. Moreover, a 10-fold cross-validation approach was adopted on 653 h of EEG. Compared with the proposed method (251 h for training, 63 h for validation and 338 h for test), the TFD-CNN-Reg has much more data (about 588 h for training, 65 h for test). In terms of the model structure, the FCN presented here is much simpler. Although TFD-CNN only has five layers, it has more parameters (44,574) compared with FCN10 (25,540) and FCN16 (44,292). Rather than extracting the complex time-frequency distribution (TFD) features, the FCN16 directly used the raw EEG as the input. A higher MCC and 95% CI were still achieved. A 23.65% $(86.09-82.8)/(100-86.09)$ relative performance improvement in test accuracy was obtained compared with the TFD-CNN-Reg method.

**Table 3.** Comparison between different methods' configurations.

|  | Dev. Set | Test Set | Dev. Method | Input |
|---|---|---|---|---|
| TFD-CNN [19] | 54 neonates (Cork) | ANSeR1 | Leave one out | TFD Features |
| TFD-CNN-Reg [20] | ANSeR1&2 | ANSeR1&2 | 10-fold CV | TFD Features |
| Proposed | ANSeR2 | ANSeR1 | Internal CV | Raw EEG |

**Table 4.** Test accuracy, confidence interval, and MCC comparison between different methods (the ensembled FCN16 prediction took the majority vote of 8 trained networks).

|  | TFD-CNN | TFD-CNN-Reg | FCN16(Ensembled) |
|---|---|---|---|
| Accuracy | 69.5% | 82.8% | **86.09**% |
| 95%CI | 65.3–73.6% | 80.5–85.2% | **82.41–89.78**% |
| MCC | - | 0.722 | **0.7691** |

### 4.2. Receptive Field

The receptive field refers to how many samples one element of a particular feature map can see from the input. A wider receptive field means that the model makes its decision on

a longer window of the time series. The receptive field can be increased either by using a deeper network or by employing longer strides in the pooling layers.

Figure 8 illustrates the receptive field of the convolutional layer in the classifier. The decisions made after the final pooling layer simply combine the outputs of the previous feature maps, yielding an overall decision which is based on approximately 60 s of EEG. However, these constituent feature maps may each have a receptive field which is significantly smaller than 60 s. We will concentrate on these as they are trainable, meaning, their receptive fields can be modified in depth, stride, etc.

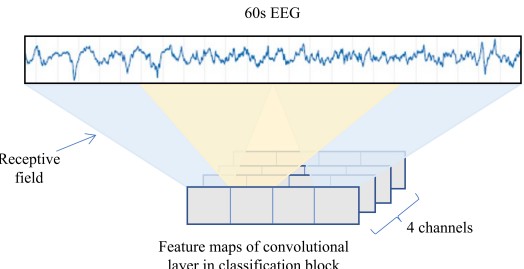

**Figure 8.** Illustration of the receptive field of the convolutional layer in the classification block.

Table 5 presents the receptive field of the three network architectures and the resulting accuracy (Acc.) and AUC (each averaged over the eight trained models, with eight initializations) for both the validation (Val.) and test sets. Surprisingly, the increase in the receptive field did not yield a large performance improvement. Although FCN16 has many more parameters and a wider receptive field, it is only slightly better in performance than the others. This means that a receptive field of approximately 30 s is large enough to extract details of the EEG to make decisions. It can also be noted that the test and validation performance across all models are quite similar, which could suggest that the proposed FCN model performance is very stable.

**Table 5.** Test accuracy, confidence interval, and MCC comparison between different classifiers (the prediction used the majority vote of 8 trained networks).

|  | Receptive Field | No. of Params | Test Acc. | True Negative Rate | Test AUC | Val. Acc. | Val. AUC |
|---|---|---|---|---|---|---|---|
| FCN10 | 29.66 s | 25,540 | 0.8458 | 0.9348 | 0.8693 | 0.8005 | 0.8674 |
| FCN13 | 39.94 s | 34,916 | 0.8421 | 0.9360 | 0.8643 | 0.7917 | 0.8689 |
| FCN16 | 49.25 s | 44,292 | 0.8572 | 0.9387 | 0.8620 | 0.8199 | 0.8639 |

Models with different depths but similar receptive fields demonstrated a similar performance. Another experiment was implemented to explore how the receptive field over a wide range affects the performance. Figure 9 illustrates the model structure of FCN13_30/60/90s. The three models configuration can be seen in Table 6. They have the same depth but different input lengths. EEG segments sampled at 16 Hz of length 30 s, 60 s, and 90 s were used as input in order to speed up the training. As they have the same depth, the three models each have the same number of parameters. Table 7 shows their receptive field and the average test, as well as the validation performance over the eight random initializations. It can be seen that, the wider the receptive field, the better the model performance will be. However, compared with the difference between the FCN13_30s and FCN13_60s models' performance, the difference between the 60 s and 90 s models is smaller, which can also be seen in the test accuracy distribution in Figure 10. Although FCN13_90s has higher mean and median values, FCN13_60s has the smallest spread. It confirms again that there is only a marginal improvement in increasing the receptive field beyond 30 s, which appears to be long enough to capture the key information.

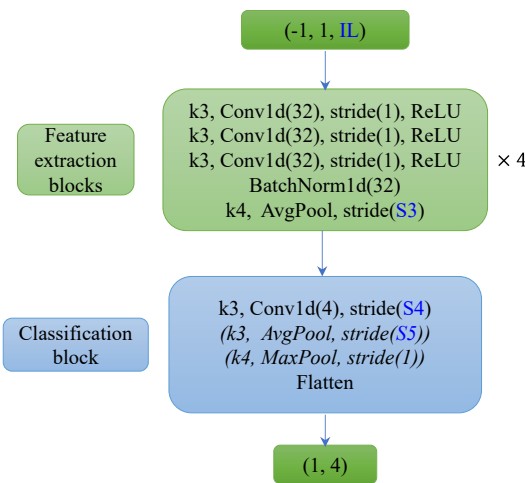

**Figure 9.** FCN13_30/60/90s (16 Hz) model architecture. The three models have the same depth (13 layers) but different input lengths (IL: 30 s, 60 s, and 90 s of EEG). The FCN13_90s model does not have the Average Pooling and Max Pooling layers in the classification block, and the FCN13_60s model only includes the convolutional layer and an Average Pooling layer.

**Table 6.** FCN13_30/60/90s model configuration. IL is the input length; S3 is stride of the average pooling layer in the feature extraction blocks; S4 and S5 are the stride of the convolutional layer and the stride of the average pooling layer in the classification block, respectively.

| | IL | S3 | S4 | S5 |
|---|---|---|---|---|
| FCN13_30s | 480 | 2 | 2 | 2 |
| FCN13_60s | 960 | 3 | 2 | 1 |
| FCN13_90s | 1440 | 4 | 1 | - |

**Table 7.** FCN13_30/60/90s model receptive field and performance comparison (averaged on 8 times random initialization).

| | Receptive Field | MCC | Test Acc. | Test AUC | Val. Acc. | Val. AUC |
|---|---|---|---|---|---|---|
| FCN13_30s | 10.5 s | 0.7261 | 0.8336 | 0.8784 | 0.6747 | 0.8609 |
| FCN13_60s | 32.69 s | 0.7692 | 0.8539 | 0.8694 | 0.7193 | 0.8693 |
| FCN13_90s | 79.88 s | 0.7773 | 0.8661 | 0.8793 | 0.7529 | 0.8893 |

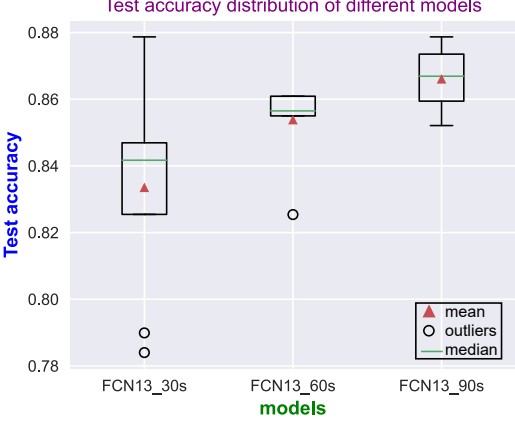

**Figure 10.** Test accuracy distribution of FCN13_30/60/90s (16 Hz) model on ANSeR1 dataset. Each model was trained with 8 different random initializations.

### 4.3. UMAP Visualization

Figure 11 shows the FCN13_90s (16 Hz) model UMAP projection from the output of the third and the fourth feature extraction blocks and from the final convolutional layer in the classification block. It can be seen that features from deeper layers present a clearer separation of the four grades. This also confirms that the model has learned more discriminatory features in the deeper layers.

Six different FCN UMAP projections can be seen in Figure 12, where features from the final convolutional layer in the classification block were used as the input of the UMAP. It can be seen that all six model UMAP projections present a good separation of the four classes. In this experiment, the embedding dimension was 2. It is important to note that the cluster size and the distance between different classes mean nothing, as UMAP uses a local distance to construct the high-dimensional graph. Unlike PCA which uses a linear operation, UMAP and t-SNE utilize a nonlinear graph-based algorithm, which makes the distance interpretation more challenging [38,39].

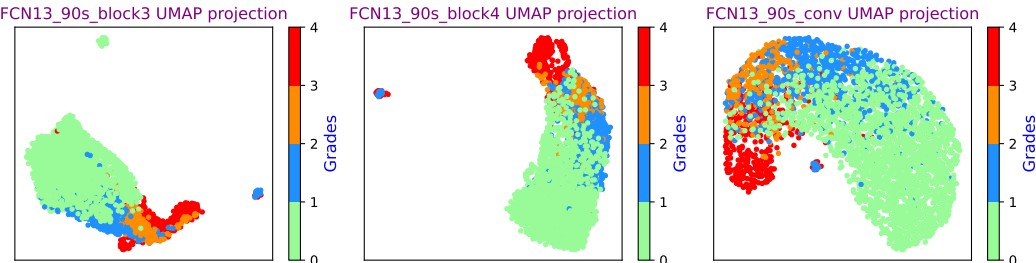

**Figure 11.** FCN13_90s (16 Hz) model with the third and fourth feature extraction blocks and the final convolutional layer output UMAP projection.

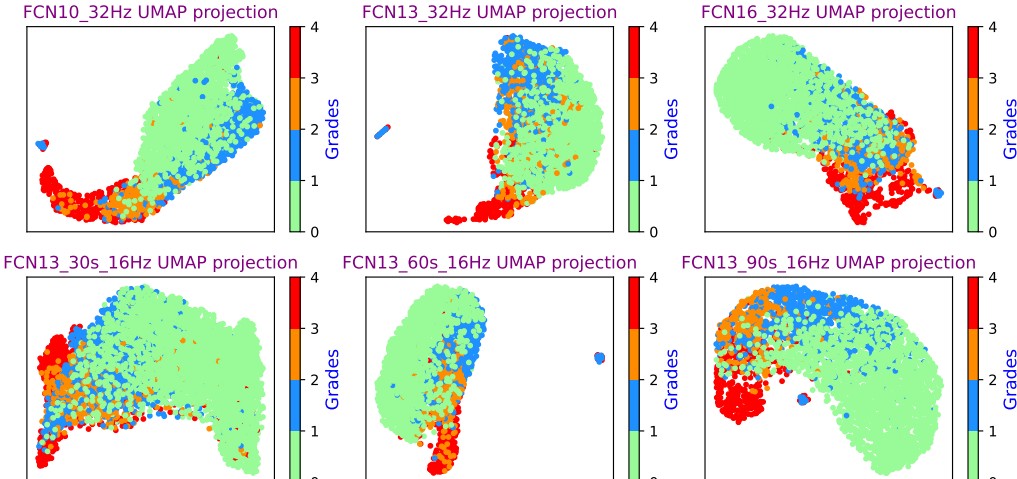

**Figure 12.** FCN UMAP projection (from the final convolutional layer output) for a range of networks.

### 4.4. UMAP Dimension Reduction

In order to test the low-dimensional representation performance of UMAP, the output of four FCN feature extraction blocks was fed into UMAP. The 96-dimensional features were projected down to a 20-dimensional representation. A hybrid classifier [40] was used with a random forest classifier to demonstrate the perrformance of the FCN/UMAP feature extraction block, as shown in Figure 13. The whole ANSeR2 dataset was used for training and the whole ANSeR1 dataset was used for testing. An 87.99% test AUC was obtained, which achieved comparable performance with the FCN classifier. This demonstrates that the FCN/UMAP combination can provide suitable discriminative EEG features. Likewise, this hybrid classification method using a pretrained FCN with UMAP as feature extraction

feeding a random forest classifier is easy to retrain as more data becomes available, or it may be possibly employed for other EEG classification problems.

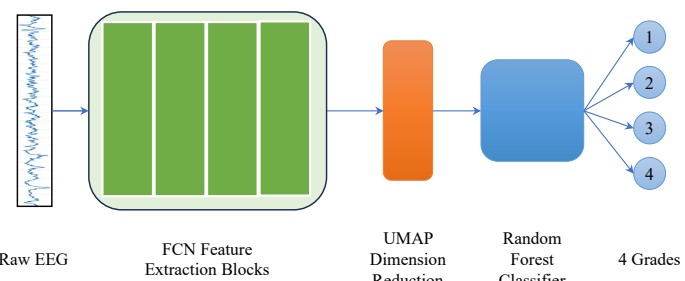

**Figure 13.** UMAP dimension reduction for further classification. The feature extraction blocks of FCN13_90s (the classification block has been removed) were used to extract features from 90 s raw EEG segment, which produced a 96-dimensional feature. UMAP embedded it as 20-dimensional for further random forest classification.

*4.5. Weak Labels*

The 1 h epoch of EEG only has one label (global label), which was assigned to all of its constituent channels and to each 60 s segment; this can effectively be thought of as weak labeling of the dataset. Each segment was treated as an independent sample during the training and testing process. As a matter of fact, not all weak labels are correct. Neurophysiologists annotated the 1 h EEG based on the whole 1 h multi-channel EEG trace. During the one hour, for example, the HIE might have changed from grade 1 to 2 and then back to 1 again. However, if most of the time, it is grade 1, the global annotation would be grade 1. This is the reason majority voting was adopted in the post-processing process. Figure 14 presents the color map of a prediction example over one hour of eight-channel EEG which has been graded as grade 2 (Moderate abnormalities) by the expert. Although many of the segments were marked as grade 3, the majority vote across all channels and time clearly provides the correct estimate of the HIE grade as 2 (Moderate). As a matter of fact, weak labels are a sort of data augmentation, which not only inject some noise but also increase the number of training samples. In [41], three kinds of data augmentation methods were used and evaluated on three different EEG-based classification tasks, which consistently produced performance improvement.

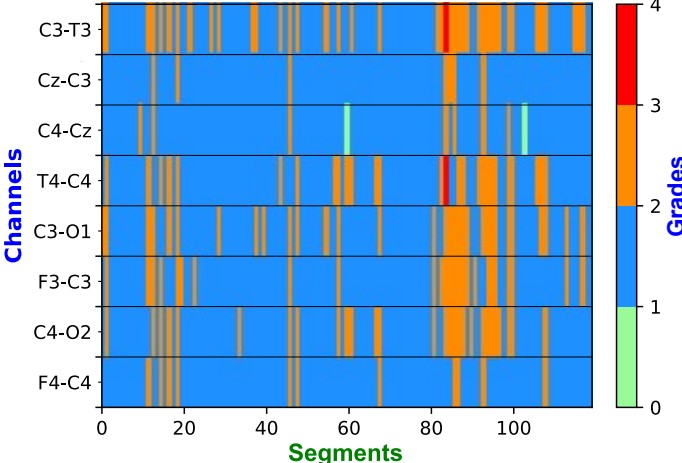

**Figure 14.** Example of the color map over the 1 h epoch showing the grading decisions for 119 segments per channel (60 s for each segment with 50% of overlap). This one hour of EEG was expertly graded as grade 2. After post-processing, the decision of the classifier is also grade 2.

### 4.6. Clinical Implementation

For clinicians, it would be helpful that the model prediction can be clearly displayed over time rather than only providing a final decision. This could allow clinicians to monitor the evolution of the injury over time. Figure 15 shows how the probabilities of the four classes of HIE injury grades evolve over one hour of multi-channel EEG. For each 90 s segment, the FCN classifier produced prediction probabilities for the four classes, which were averaged over the eight channels. From Figure 15, it can be seen that, most of the time, grade 1 had the highest probability, but at around 50 min it deteriorated into grade 2 and 3. Subsequently, it gradually went back to grade 1. Finally, the majority vote correctly predicted it as grade 1.

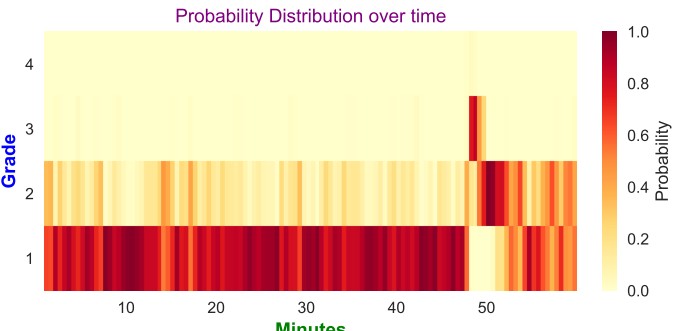

**Figure 15.** Example of the probability distribution of the 4 grades (averaged over 8 channels). The final prediction is grade 1, which is the same as the expert's annotation. The darker color indicates a higher probability is predicted for the corresponding grade, whereas the lighter color means the probability is lower.

In some cases, not all of the eight channels are available or indeed it may not be possible to apply all the electrodes to the baby's scalp within the first critical 6 h. HIE is a global injury and could possibly be graded with one channel of EEG, which is easier to set up in the NICU within the first 6 h. It is therefore useful to determine the performance of each single channel of EEG. Figure 16 shows the test accuracy of using different channels based on the whole ANSeR1 dataset. The final decision was obtained through a majority vote method. If only one channel was used, the majority vote was processed only in the time dimension rather than across time and channels. As expected, when all of the channels were considered, the test accuracy was the highest (87.57%). However, it can also be noted that the test accuracy using a single channel is only moderately reduced. In making decisions only based on one channel, the best performance was only reduced by 2.63% (85.21% for channel "C3-T3") compared with all channels. Even using the lowest one ("C4-Cz"), it also achieved 81.66% test accuracy. This confirms that it is possible to grade the injury only based on one channel. The accuracy loss only ranges from 2.63% to 5.91%. However, it should be acknowledged that multiple channels provide some robustness to artefacts and detached electrodes.

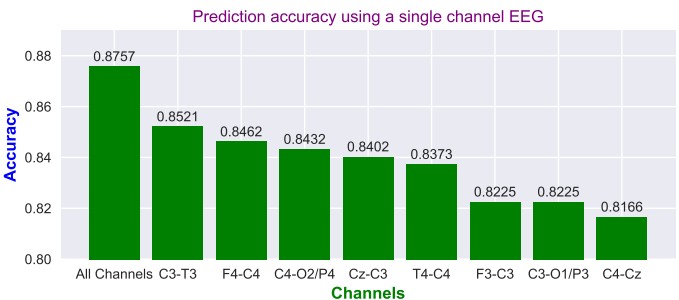

**Figure 16.** Prediction accuracy using different channels based on the majority vote over time and tested on the whole ANSeR1 dataset.

## 5. Conclusions

This paper presented a fully convolutional neural network architecture for HIE grading. The model was developed and assessed on two large HIE datasets containing unedited continuous 1 h segments of multi-channel EEG, where there was a substantial mismatch between training and testing conditions. The training process was based on 60 s segments with weak labels. The channel independence and majority voting made the prediction more robust. The minimally pre-processed EEG time-series data can be directly inputted without the need for feature engineering. The raw EEG was only downsampled, filtered, and scaled, and no other artefact removal or normalization procedures were used. Compared with the state-of-the-art, this architecture is simpler and deeper, but with fewer parameters. Compared with the TFD-CNN-Reg method [20], it was trained on a smaller dataset but achieved comparable performance with a much larger, completely different test set.

A powerful dimension reduction technique, UMAP, was used to visualize the FCN feature extraction block output, which has demonstrated a clear separation of the four classes on two-dimensional space. A 20-dimensional feature vector embedded from UMAP and FCN feature extraction blocks was fed into a simple random forest classifier to present the feature extraction performance of the FCN/UMAP combination. The performance obtained was comparable with the full FCN classifier.

A clinical implementation of the post-processing was performed, where the model prediction result was displayed over time rather than only providing a final grade prediction. In addition, as HIE is a global injury, it is feasible to make decisions only based on one channel of EEG with an accuracy loss of 2.63–5.91%.

**Author Contributions:** Conceptualization, S.Y. and G.L.; methodology, S.Y. and G.L.; software, S.Y.; validation, S.Y., G.L. and W.P.M.; formal analysis, S.Y. and G.L.; investigation, S.Y., G.L. and W.P.M.; resources, W.P.M. and G.B.B.; data curation, W.P.M. and G.B.B.; writing—original draft preparation, S.Y.; writing—review and editing, S.Y. and G.L.; visualization, S.Y.; supervision, G.L. and W.P.M.; project administration, G.L. and W.P.M.; funding acquisition, W.P.M. and G.B.B. All authors have read and agreed to the published version of the manuscript.

**Funding:** This work was supported by Science Foundation Ireland (SFI) (19/FFP/6782). This study was also supported by a Strategic Translational Award and an Innovator Award from the Wellcome Trust (098983 & 209325).

**Institutional Review Board Statement:** The study was conducted in accordance with the Declaration of Helsinki. The data used in this work was obtained from infants recruited for two prospective, multicenter, cohort studies (ClinicalTrials.gov Identifier: NCT02160171 and NCT02431780) from eight tertiary-level neonatal intensive care units across Europe (Ireland, United Kingdom, Sweden, The Netherlands) Ethics committee approval from each recruiting site was obtained before recruitment commenced. This work was approved by the Clinical Research Ethics Committee of the Cork Teaching Hospitals (Reference code of ECM 5 (8) 14/10/14 & ECM 3 (nnnn) and date of 22/02/2022 approval).

**Informed Consent Statement:** Informed consent was obtained from all subjects involved in the study.

**Data Availability Statement:** A part of this dataset is publicly available: https://zenodo.org/records/6587973#.YzriO3aZOUk, accessed on 21 September 2023. For the rest of the data, the authors do not have the permission to share.

**Acknowledgments:** We would like to thank the ANSeR consortium: INFANT Research Centre and Department of Paediatrics and Child Health, University College Cork, Cork, Ireland; Institute for Women's Health, University College London, London, UK; Utrecht Brain Center, University Medical Center Utrecht, Utrecht University, The Netherlands; Department of Neonatal Medicine and Division of Paediatrics, Department CLINTEC, Karolinska University Hospital, Karolinska Institutet, Stockholm, Sweden; Rotunda Hospital, Dublin, Ireland; Royal London Hospital and Queen Mary University of London, London, UK; Department of Clinical Neurophysiology, Great Ormond Street Hospital for Children NHS Trust, London, UK; Homerton University Hospital NHS Foundation Trust, London, UK; Clinical Neurophysiology, University Medical Center Utrecht, Utrecht, The Netherlands.

**Conflicts of Interest:** G.B. Boylan is founder and shareholder in Kephala Ltd. and Cergenx Ltd.; has provided consultancy to GW Pharmaceuticals, Nihon Kohden and UCB Pharma.

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
