# Peer review of "Neonatal Hypoxic-Ischemic Encephalopathy Grading from Multi-Channel EEG Time-Series Data Using a Fully Convolutional Neural Network"

_technologies, doi:10.3390/technologies11060151_

Round 1
Reviewer 1 Report
Comments and Suggestions for Authors
This paper describes raw EEG analysis as a predictor for the diagnosis of a neonatal hypoxic-ischemic encephalopathy. This is an important clinical problem. The paper is well written, skilled and informative, methods are clearly described, discussion is adequate.
There is some room for improvement:
Line 122: “In total, 472 neonates were recruited, but for the purpose of our analysis, only 284 infants with HIE were included.” Why? How did the authors select this sample?
Line 143 and following: exact definition of inter-burst interval in terms of duration and amplitude?
Line 212: The authors might explain the advantages of the majority vote procedure and the reason why they used this procedure
Table 2: It is not enough to report only sensitivity and PPV, true negatives and specificity should be given, too.
Table 5: TN (true negatives) should be reported, too. An overall accuracy of 86% is not high, if relevant clinical decisions depend on that information.
Fig 14: Which are the predictors for the random forest classifier? After all: Which are the final EEG predictors identified by FCN?
Line 421: one channel classification. One channel recordings are prone to artefacts. It is unjustifiable to rely on one channel EEG recordings if relevant decisions depend on the recording. Additionally, this conclusion is a logical vicious circle: if one treats (line 176) every sample as an image of a single channel, the information about the location gets lost.
Reviewer 2 Report
Comments and Suggestions for Authors
The authors suggest employing a fully convolutional neural network for HIE grading in neonates using direct EEG signals. The approach is highly intriguing and shows promise as the proposed model achieves state-of-the-art accuracy with a simpler architecture and fewer parameters. I have a few minor suggestions:
Please replace the '~' symbol, which typically indicates approximation, with '-' to represent correct intervals.
I noticed that Figure 12 and Figure 13 seem to contain empty graphs. Could you please review and update these figures to include the necessary content?"
